# Research on Seismic Wave Quality Factor of Marble Jointed Rock Mass under SHPB Impact

**Changkun Sun [1,2,*], Changhong Li [1] and Xiaoming Wei [3]**

[1] School of Civil and Resource Engineering, University of Science and Technology Beijing, Beijing 100083, China
[2] Yunnan Gold Mining Group Co., Ltd., Kunming 650299, China
[3] BGRIMM Technology Group, Beijing 100160, China
[*] Correspondence: d202110007@xs.ustb.edu.cn

**Abstract:** In order to quantitatively describe the energy dissipation law of jointed rock mass, we obtained the jointed cores in laboratory conditions using marble from the roof and floor of Jinchanghe Lead–zinc mine in Baoshan. The dissipative degree of stress wave in marble jointed rock mass is measured by introducing quality factor $Qs$ parameter. Based on the experimental principle of Split Hopkinson Pressure rod loading device (SHPB), we proposed a three-wave energy method of incident wave, reflected wave and projected wave for calculating jointed rock samples' quality factor $Qs$ based on stress wave energy. Using the SHPB test system for multiple specimens taken from the same piece of rock mass shock compression experiment, the three groups of specimens under different loading conditions gained incident wave and reflected wave and transmission wave experimental data, using the method of stress wave energy to deal with stress wave data, and calculate the joint sample maximum storage energy, dissipation energy and $Qs$ quality factors. The results show that: ① The non-destructive breaking time–history strain of Dali rock mass under impact load is obtained by SHPB dynamic test system; combined with the deformation energy and dissipation energy calculation principle of quality factor, six groups of $Qs$ experimental values are obtained. Although the $Qs$ experimental values are discrete, the overall deviation is small with an average of 43.07. ② AUTODYN-Code was used to simulate the damage and fracture characteristics of rock mass with different quality factors under explosive dynamic load. The results showed that the radius of rock mass compression shear damage area gradually increased with the increase in porosity, but it was not obvious.

**Keywords:** energy dissipation; quality factor $Q_s$; SHPB impact test; numerical simulation





## 1. Introduction

The rock mass structure is composed of many discontinuities, such as joints, faults and other weakening surfaces, which play a major role in controlling the mechanical properties of the rock mass structure [1]. These discontinuities will block the propagation of stress wave in rock mass, slowing down and weakening the propagation of stress waves [2–4]. In mining engineering, the rock mass is unavoidably disturbed by frequent blasting shock waves, which gradually damages and fractures rock mass under repeated disturbances and eventually leads to rock mass destruction, seriously threatening the safety of mining [5–11]. Therefore, it is of great significance to study energy dissipation of jointed rock mass and obtain seismic wave quality factors of rock mass for mine safety production. At present, most of the research focuses on the amplitude change in stress waves after they pass through the joint, and the energy attenuation of stress waves is rarely analyzed; stress wave energy attenuation is an important part of stress wave propagation law, which can provide a certain reference for blasting vibration analysis and prediction [12,13]. Therefore, it is valuable to fully understand the influence of joints on stress wave energy attenuation in rock mass.

Split Hopkinson Pressure rod (SHPB) experimental device is widely used to test dynamic mechanical properties of rock mass, and it is the most commonly used dynamic loading device. Li Xibing [14–18] established a dynamic and static combined loading device by combining the SHPB experimental device and the axial compression system, where he conducted multiple impact tests on concrete of different curing ages, analyzed the damage changes in concrete under multiple impact, and simulated typical static–dynamic combined loading problems such as instability of pillar under disturbance. SHPB tests were carried out on granite samples with circular and square holes under different axial static load and the same dynamic impact load, respectively, revealing the evolution law of pore structure and damage degradation of granite rock under dynamic and static load. Gong Fengqiang [19–21] analyzed the specimen size in an SHPB experiment and gave reasonable size suggestions. Using the modified three-dimensional Hopkinson combined loading test device, he carried out a preliminary study on the mechanical properties of sandstone. He discussed the evolution law of compressive strength of sandstone when confirming that pressure and axial pressure change each other. Based on the analysis of the influence of confining pressure on the impact deformation and strength of sandstone, the energy dissipation law and failure mode of sandstone in the impact process are explored. The results show that with the increase in confining pressure and strain rate, the compressive strength of sandstone tends to strengthen, and the critical incident energy for sandstone failure augments with the increase in confining pressure. There is a linear increasing relationship between the absorption energy per unit volume and strain rate, and the degree grows with the increase in confining pressure. Xu Songlin [22,23] reformed the SHPB experimental device and used a square rod to load the specimen in three directions, realizing the dynamic impact test of rock mass under three-dimensional stress. Hao Zhaobing [24] elaborated various attenuation mechanisms, measurement methods of quality factor $Q$ and main influencing factors from the aspect of rock physics. Kong Xiangzhen [25] researched that the nonlocal model based on damage can solve all the limitations of the original nonlocal model. Mohammad Reza Khosravani [26] used the Holmquist–Johnson–Cook constitutive model to characterize the dynamic behavior of ultrahigh-performance concrete, which is a new generation of concrete with higher strength compared with traditional concrete. The reflected and transmitted waves obtained from experiments are used as the input in the inversion process. F.LI [27] researched that the increase in compressive strength in SHPB test was largely caused by the change in stress state in the sample; that is, when the strain rate was greater than the transition strain rate, the transverse limit was introduced, from uniaxial stress state to multiaxial stress state. Wang C L et al. [28,29] combined this with the deformation characteristics of granite, and acoustic emission (AE) technology was well applied in revealing the evolution law of micro-cracks in the process of rockburst. Chen S J et al. [30,31] established the elastic–plastic-brittle mutation rockburst model of coal rock with structural surface, introduced the concept of the coal rock volume energy potential function, and researched the relationship between energy accumulation and dissipation during the coal rock dynamic deformation and rupture process to obtain the rockburst energy condition.

In this paper, to control blasting vibration and prevent its threat to adjacent structures, it is necessary to fully understand the propagation law of stress wave in rock mass. The marble from the roof and floor of Jinchanghe Lead–zinc mine in Baoshan was used as test material. The SHPB test system was used to conduct impact compression tests on several specimens from the same rock mass. The quality factor $Qs$ was introduced to measure the dissipative degree of stress wave in marble jointed rock mass. Combined with the calculation principle of deformation energy and dissipation energy of quality factor, six groups of $Qs$ experimental values were obtained. AUTODYN-Code was used to simulate the damage and fracture characteristics of rock mass with different quality factors under explosive dynamic loading.

## 2. SHPB Test System

### 2.1. Test System

The SHPB test system consists of a rod system, a base system, a power system, a buffer, and a data acquisition system, as shown in Figure 1a.

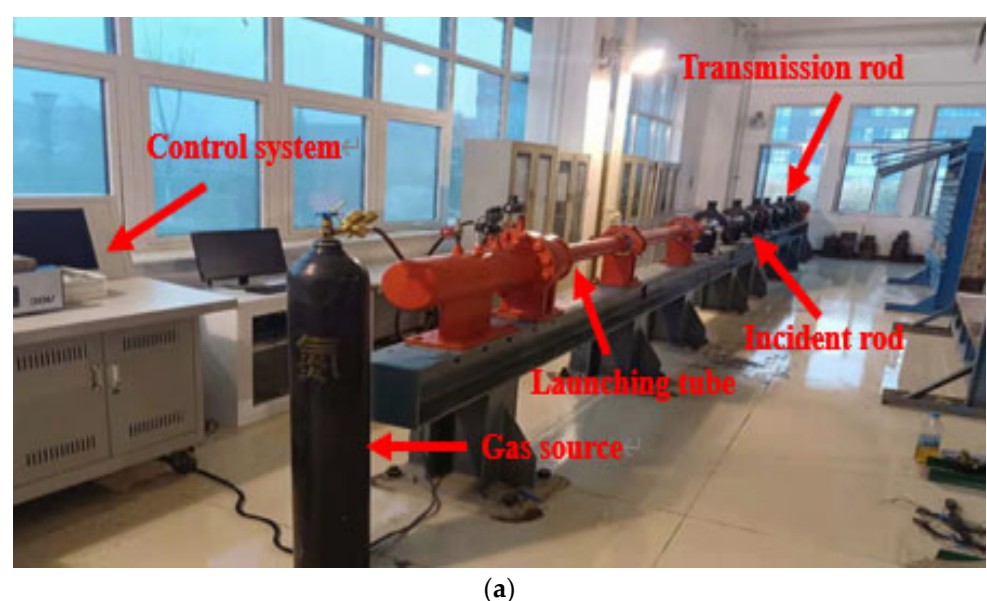

(**a**)

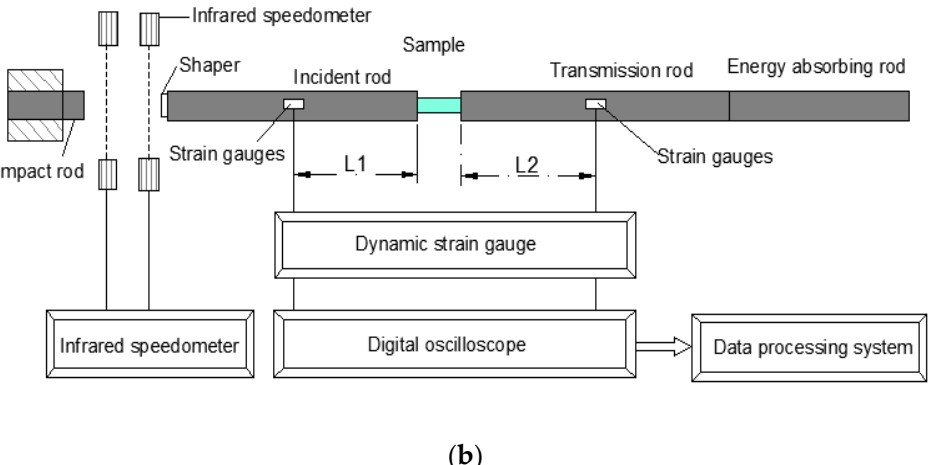

(**b**)

**Figure 1.** SHPB loading device and schematic diagram of dynamic test system. (**a**) SHPB loading device, (**b**) Schematic diagram of dynamic test system.

The relevant parameters of SHPB rod systems are as follows: The elastic rod is made of high strength spring steel, with a density of 7600 kg/m$^3$, an elastic modulus of 210 GPa, a yield strength of 355 GPa, and a Poisson's ratio of 0.33. The diameter of the elastic rod is 80 mm, the length of the incident rod is 4000 mm, the length of the transmission rod is 4000 mm, the length of the absorption rod is 1000 mm, and the measured wave velocity is 5136 m/s. The shell size is $\phi$ 80 × 500 mm, and its parameters are the same as the incident and transmission rods. The power system consists of a high-pressure nitrogen cylinder, a pressure storage chamber and a control device. The test strain gauge SG of the incident rod is pasted on the end L1 of the incident rod, and the strain gauge SG of the transmission rod is pasted on the front L2 of the transmission rod, where L1 = L2 = 1.5 m. The impact velocity of the impact rod was measured by the photoelectric velocimetry device, and the strain signals on the incident rod and transmission rod were collected by the ultra-dynamic strain meter. Meanwhile, the voltage signals collected were output by the acquisition system

and converted into corresponding data information through theoretical calculation. The schematic diagram of the dynamic test system is shown in Figure 1b.

### 2.2. SHPB Test Principle

The power system is controlled by the built-in software of the computer, and a pressure threshold is set for the pressure chamber. After the pressure reaches a predetermined threshold, the projectile will be pushed forward. When the projectile is accelerated by high-pressure nitrogen inside the launching tube, it impacts the incident rod along the axial direction at a certain outlet speed. After the projectile is discharged from the chamber, the velocity of the projectile will be recorded by the infrared tachometer. A compressive stress wave will be generated in the incident rod when it is impacted. At this time, the incident rod will undergo one-dimensional elastic deformation, and a stress pulse will be formed in the rod and propagate forward along the incident rod. When the stress pulse propagates to the interface I–I between the specimen and the incident rod, the reflected tension unloading wave is formed at the interface because the incident rod wave impedance $\rho_1 v_1$ is larger than the specimen wave impedance $\rho_1 v_1$. The reflected tension wave propagates back into the incident rod and forms a transmitted compression wave into the specimen at the same time. Because the parameters of the rock specimen are different from those of the incident rod, the stress wave transmitted into the specimen does not propagate in the form of elastic wave but carries on energy transfer in the form of stress wave disturbance. After the compression wave propagates a specimen length in the specimen, it will be reflected again at the interface II–II between the specimen and the transmission rod to form the reflected compression wave into the specimen. The compression wave will be reflected and transmitted back and forth between the interfaces I–I and II–II, and finally reach the stress balance. In addition, a compressed transmitted wave will also form at the interface of II–II and enter the transmission rod, then enter the absorption rod and buffer through the projection rod and finally return to dissipation. The wave conversion at the interfaces of I–I and II–II is shown in Figure 2.

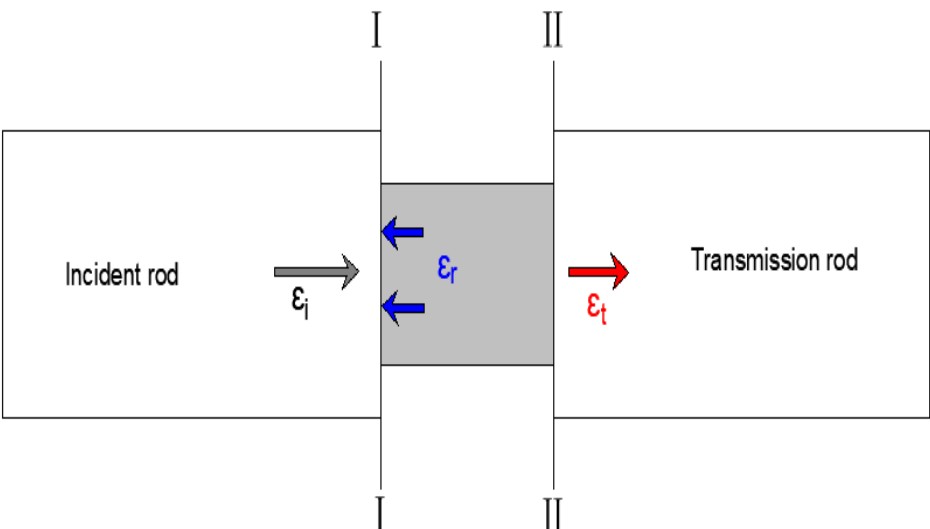

**Figure 2.** SHPB test principle.

According to the stress wave theory and one-dimensional wave hypothesis, if we ignore the dispersion effect of stress wave propagation in elastic rod, then the propagation of stress wave in elastic rod has no distortion and no attenuation. According to the displacement continuity, the compression pulse in the incident rod is called $\varepsilon_i$, the reflected wave generated at the interface between the incident rod and the specimen I–I is regarded as $\varepsilon_r$, and the transmitted wave generated at the interface between the transmission rod and the specimen II–II is regarded as $\varepsilon_t$. Based on the above analysis, the strain rate, strain

and average stress generated in the specimen at time T can be obtained, and the function expression is shown as follows.

$$\dot{\varepsilon}(t) = \frac{C_0}{L}[\varepsilon_i(t) - \varepsilon_r(t) - \varepsilon_t(t)] \tag{1}$$

$$\varepsilon(t) = \frac{C_0}{L} \int_0^t [\varepsilon_i(t) - \varepsilon_r(t) - \varepsilon_t(t)]dt \tag{2}$$

$$\sigma(t) = \frac{A_0 E_0}{2A}[\varepsilon_i(t) - \varepsilon_r(t) - \varepsilon_t(t)] \tag{3}$$

where $L$ is the specimen length; $C_0$ is the elastic wave velocity of the incident rod and transmission rod; $E_0$ is the elastic modulus of the incident rod and transmission rod; $A$ is the cross-sectional area of the specimen.

The three waveforms $\varepsilon_i(t)$, $\varepsilon_r(t)$ and $\varepsilon_t(t)$ can be measured by SHPB system, and $\dot{\varepsilon}(t)$, $\varepsilon(t)$, $\sigma(t)$ can be obtained by combining the above three Equation at the same time, which is commonly known as "three-wave method".

### 2.3. Experimental Con-Figuration and Specimen Preparation

Firstly, a number of $\phi$ 50 × 100 mm specimens were obtained in the same marble rock mass by using a NUMERICAL control drilling corer. Secondly, it was cut by a large cutting machine and segmented into several specimens of moderate size. Finally, the specimens were polished by a CNC double-end grinding machine. After grinding, the size of the specimen was $\phi$ 50 × 35 mm. Meanwhile, the flatness of the specimen end face and the parallelism of the double end faces were ensured to meet the relevant standards in *Standard for Experimental Methods of Engineering Rock Mass* (GB/T 50266-2013), that is, the error of the flatness of the end face is less than 0.01 mm. The processing system is shown in Figure 3.

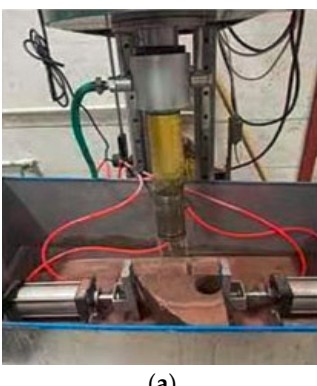 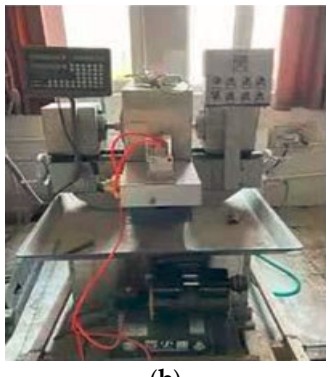 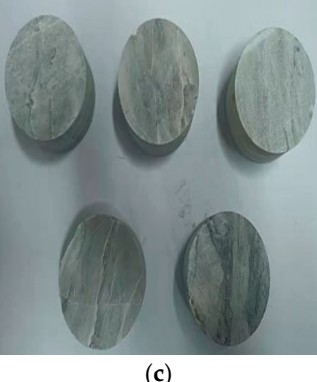

(**a**)  (**b**)  (**c**)

**Figure 3.** Specimen processing equipment and specimen preparation. (**a**) CNC core machine; (**b**) CNC grinding machine; (**c**) Prepared specimen.

### 2.4. The Experimental Results

The parameters of the processed specimens were measured by the sound velocimeter, and the compression wave velocity Cp = 3731 m/s and shear wave velocity Cs = 2308 m/s were obtained.

## 3. SHPB Dynamic Test Experiment

### 3.1. Experimental Con-Figuration and Specimen Preparation

Before the SHPB acquisition experiment of seismic wave quality factor of jointed rock mass is carried out, the SHPB system should be tested by air strike without specimen, and a stable half-sine incident wave is obtained, and the energy of incident wave is not enough to damage the specimen. The debugging steps are as follows:

(1) Adjust the support of the rod system and restrain the adjustable center frame of the elastic rod, so that the central axis of the projectile, the incident rod, the transmission rod

and the absorption rod are at the same level, and at the same time it is necessary to ensure that the incident rod and the transmission rod end face have a high degree of coincidence;

(2) The radial and annular strain gauges are, respectively, pasted at the designated positions of the incident and transmission rods, and the shielded acquisition line is connected. The glue is fully solidified by standing still for 24 h. The strain gauges and rods are firmly pasted without bubbles, as shown in Figure 4;

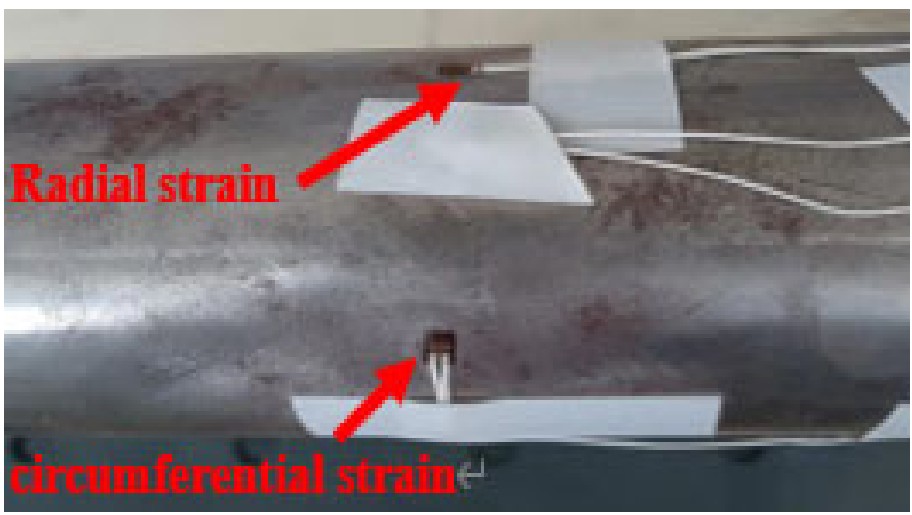

**Figure 4.** Schematic diagram of pasting radial and circumferential strain gauges.

(3) Connect the strain gauge data acquisition line, bridge box (1/4 bridge), super dynamic strain gauge channel and dynamic test system, and open the test system for preheating;

(4) Turn on the power control system, fix the distance between the shell and the gas escape port in the gun barrel at 80 mm, the nitrogen pressure in the given pressure chamber of 0.21 MPa, and obtain a relatively standard stability of half-sine incident waveform by fine-tuning the spatial position of the shell, the incident rod and the transmission rod and the parameters of the dynamic acquisition channel;

(5) Evenly cover the two end faces of the specimen with vaseline and place it on the center line of the rod body between the incident rod and transmission rod. Meanwhile, ensure that the two end faces of the specimen are in close contact with the incident rod and transmission rod, respectively, and cover with a metal safety cover as shown in Figure 4;

(6) To accurately obtain rock quality factors it is necessary to ensure that the specimen is not damaged by incident stress waves. Therefore, after the half-sinusoidal stress wave is obtained, it is necessary to adjust the stress to obtain the incident condition that can meet the experimental requirements. After continuous adjustment of cavity pressure and distance between shell and gas escape port, it is concluded that experimental requirements can be met when the above two parameters are 0.15 MPa and 60 mm, respectively. Figure 5 shows the schematic diagram of specimen loading, and Figure 6 shows part of specimen washed out during debugging.

Based on the radial and circumferential strain voltage signals obtained in Step (6), the radial and circumferential strain data of incident wave, reflected wave and transmitted wave can be obtained by calculating Equation (4).

$$\varepsilon_{r,h}(t) = \frac{4\Delta U(t)}{nEK} \tag{4}$$

In the Equation (4), $\varepsilon$ is the strain value transformed by voltage signal after calculation, $\Delta U$ is the measured voltage signal, $n$ is 1000 times gain amplification of ultradynamic strain gauge ($n = 1000$), $E$ is bridge supply voltage 2 V, $K$ is strain gauge sensitivity coefficient 2.1.

By plugging the radial and circumferential strain data calculated by Equation (4) into Equation (5), the time–history strains of incident wave, reflected wave and transmitted wave can be obtained, and the time–history curves of strain are shown in Figure 7.

$$\varepsilon_i(t),\ \varepsilon_r(t),\ \varepsilon_t(t) = \frac{1}{1-\mu_d^2}(\varepsilon_r(t)+\mu_d\varepsilon_h(t)) \tag{5}$$

where $\varepsilon_i,\ r,\ t(t)$ is the time–history value of incident wave, reflected wave and transmitted wave, respectively, and $\mu_d$ is the dynamic Poisson ratio of rock 0.26.

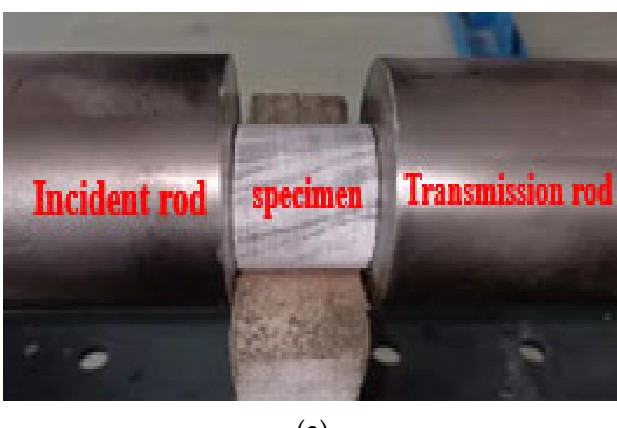

(a)

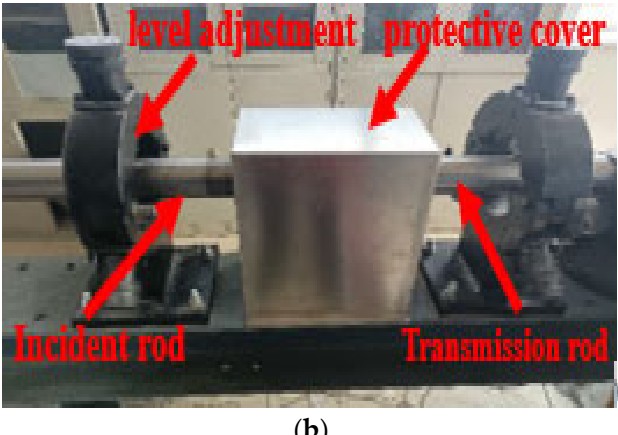

(b)

**Figure 5.** Schematic diagram of specimen loading. (**a**) Schematic diagram of loading; (**b**) Schematic diagram of loading device.

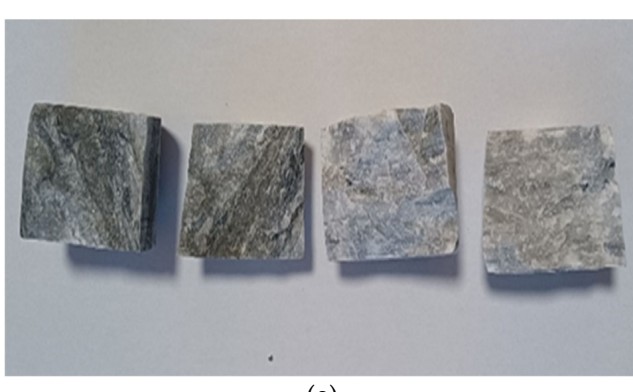

(a)

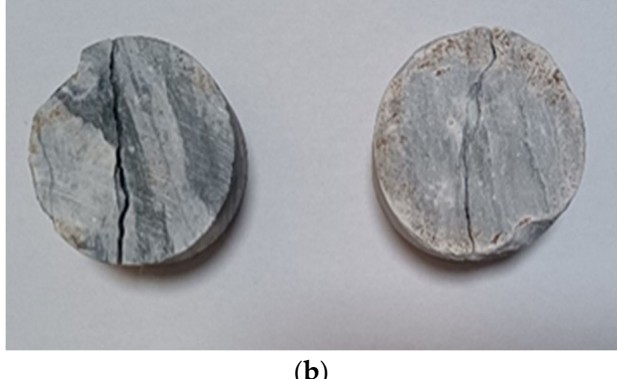

(b)

**Figure 6.** Schematic diagram of partially damaged specimens during SHPB commissioning. (**a**) Expansion diagram of test block failure; (**b**) Test block destruction diagram.

*3.2. SHPB Experiment and Test Results*

Based on the debugging results in Section 2.1, we learned that the incident conditions that can meet the experimental requirements are that the nitrogen pressure is 0.15 MPa and projectile is 60 mm away from the gas escape port. To verify whether there are mesoscopic open joints in the specimen under the above condition, a low loading rate test with increase nitrogen pressure of 0.15 MPa and the projectile 50 mm from the gas escape port is added for the same specimen. If the two test results are in good agreement, it indicates that the test piece does not produce micro open joints under the first impact, and the test conditions obtained through commissioning can meet the requirements.

Three specimens were taken under the above loading conditions: ① the nitrogen pressure was 0.15 MPa, and the projectile was 60 mm away from the gas outlet; ② nitrogen pressure 0.15 MPa, shell distance 50 mm from gas escape port were tested, respectively. The

same processing method in Section 2.1 was used to process the measured strain voltage signal, and the obtained three-wave strain time–history curve is shown in Figure 8.

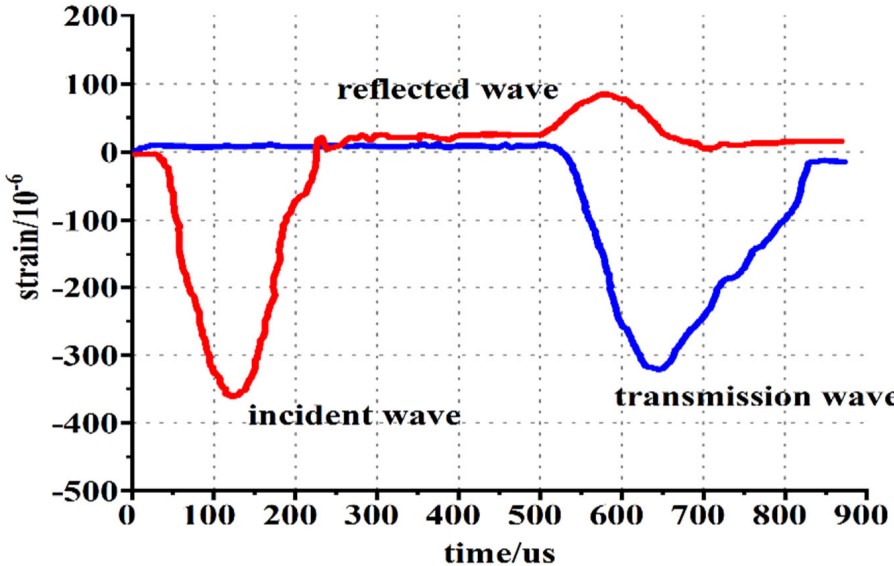

**Figure 7.** SHPB three−wave strain time−history curve.

The fundamental frequency of the specimen was obtained by using a dynamometer, and the dynamic elastic modulus of the specimen was 34.68 g by combining the fundamental frequency parameters with the wave velocity measured by a sonic meter. The dynamic Poisson's ratio $\mu_d = 0.26$ can be obtained by combining elastic wave theory with Poisson's ratio Equation.

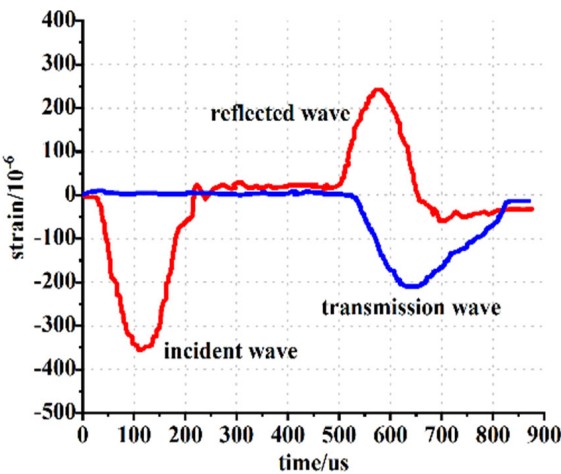
Specimen 1 under the condition of ①

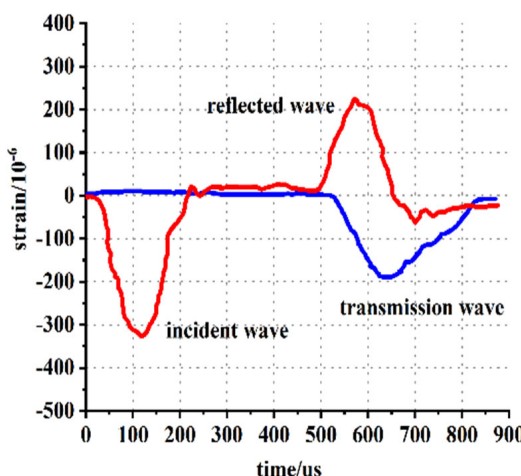
Specimen 1 under the condition of ②

**Figure 8.** *Cont.*

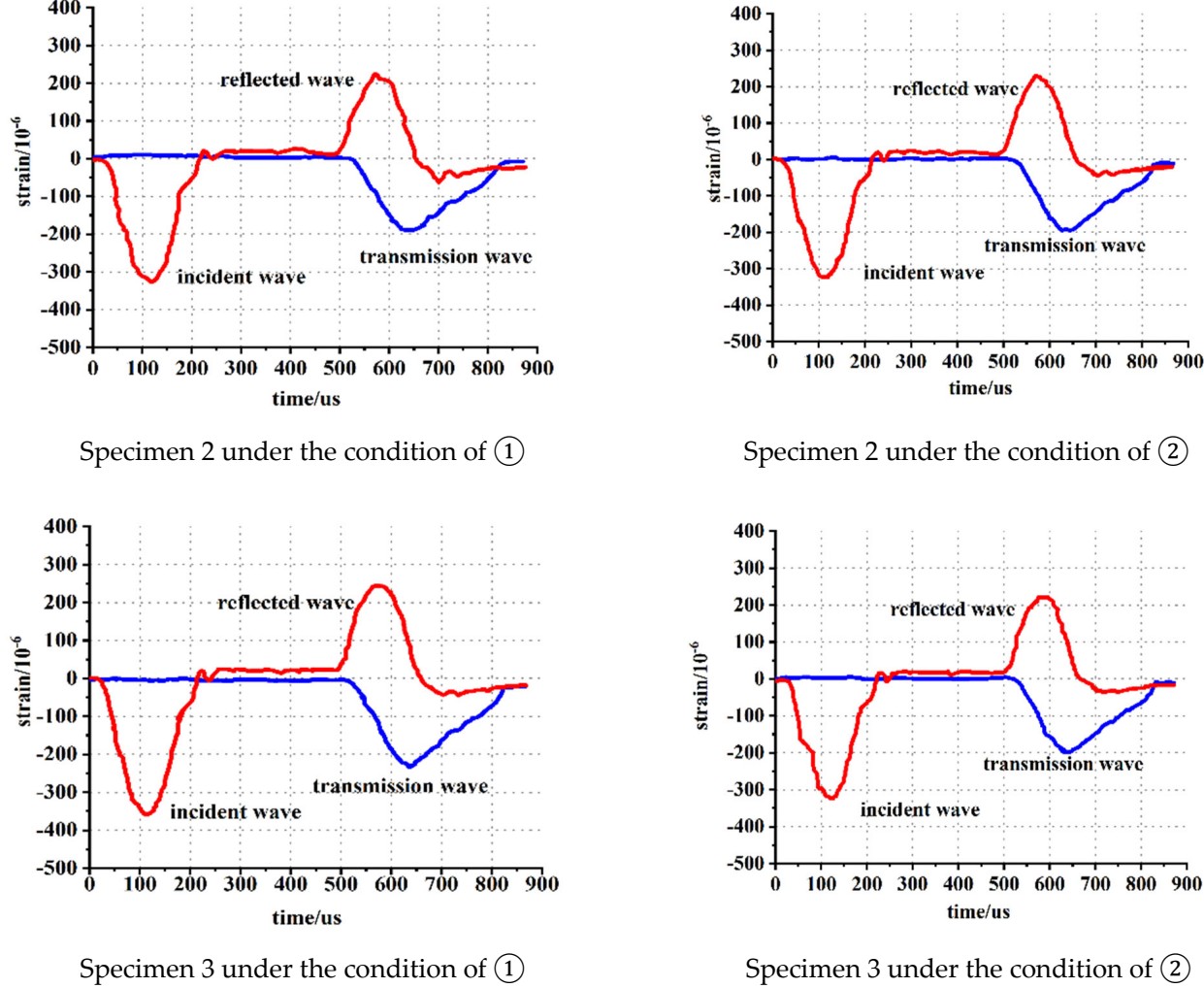

Specimen 2 under the condition of ①　　　　　　Specimen 2 under the condition of ②

Specimen 3 under the condition of ①　　　　　　Specimen 3 under the condition of ②

**Figure 8.** Time−history curves of three−wave strain of specimens under different loading conditions.

## 4. Rock Mass Seismic Wave Quality Factor

### 4.1. Quality Factor Calculation Method

Rock mass seismic wave quality factor (Shorthand for the $Q_s$, can be called quality factor) is the ratio of maximum variable performance to dissipated energy, and is a dimensionless parameter. Based on the experimental principle of the SHPB test system, it can be seen that in the test process the total energy of the separated Hopkinson pressure rod test system consists of four parts: incident wave energy $W_i$, reflected wave energy $W_r$, transmitted wave energy $W_t$, and joint sample deformation energy $W_s$. In the process of the SHPB test, the stress waves and deformation of joint samples in elastic rods such as projectile, incident rod, transmission rod and absorption rod are time dependent. Therefore, the above four energies can be expressed as time-dependent functions, namely $W_i(t)$, $W_r(t)$, $W_t(t)$ and $W_s(t)$. Under the action of incident stress wave, the incident elastic rod exerts dynamic load on the jointed rock mass sample, and the work done by the elastic rod on the jointed rock mass sample can be denoted as $U(t)$. The energy of the work done by the incident rod on the jointed rock mass sample comes from the energy of the elastic wave in the rod, so $U(t) = W_i(t) - W_r(t) - W_t(t)$. According to the first law of thermodynamics, the work done by incident elastic rod $U(t)$ is all converted into the deformation energy $W_s(t)$ of jointed rock mass sample. Therefore, the deformation energy $W_s(t)$ of the joint

sample can be obtained from the stress wave energy in the incident elastic rod and can be expressed as the Equation (6).

$$W_s(t) = U(t) = W_i(t) - W_r(t) - W_t(t)0 \ll t \ll t_0 \tag{6}$$

where t is the independent variable time, $t_0$ is the dynamic loading duration, which can be determined by the stress wave duration. According to the elastic theory, the energy of incident wave, reflected wave and transmitted wave can be calculated from the following Equation to obtain Equations (8)–(10).

$$W_i\,(t) = Ac_0E\int_0^{t_0} \varepsilon_i^2(t)dt \tag{7}$$

$$W_r\,(t) = Ac_0E\int_0^{t_0} \varepsilon_r^2(t)dt \tag{8}$$

$$W_t\,(t) = Ac_0E\int_0^{t_0} \varepsilon_t^2(t)dt \tag{9}$$

Substitute Equations (7)–(9) into Equation (6) to get Equation (10).

$$W_s\,(t) = Ac_0E\int_0^{t_0} \varepsilon_i^2(t) - \varepsilon_r^2(t)dt - \varepsilon_t^2(t) \tag{10}$$

According to Equation (10), the stress wave data measured by SHPB experiment can be used to calculate the deformation energy of jointed rock samples.

According to theoretical analysis and Equation (10), the deformation energy $W_s(t)$ of jointed rock mass sample in the SHPB experiment is not a fixed value, but a quantity that changes with loading time. Under the action of stress wave in the incident rod, joint rock specimens under the action of stress wave deformation will occur. As stress wave energy is passed to the joint rock sample, sample deformation energy will also increase gradually, and reach the maximum value, at this time, the maximum stored energy is $W_s max$. Then, with the continuous dynamic loading, the stress acting on the jointed rock samples will gradually decrease. The deformation of the sample gradually recovers, and the deformation energy released in the process of deformation recovery is transferred to the elastic rod in the form of elastic wave. The deformation energy of the sample gradually decreases until the end of dynamic loading. Due to the inelastic characteristics of jointed rock samples, in the process of dynamic loading and unloading part of the deformation energy will be consumed by jointed rock samples, resulting in a loss of stress wave energy; this part of the loss energy is equal to the energy consumed by the rock samples, which is called dissipation energy $W_l$. Because of the inelastic characteristics of jointed rock mass samples, $W_l$ is consumed in the loading–unloading process and will no longer transfer the elastic rod. The other part of the deformation energy is removed, which is transferred back to the elastic rod in the form of stress wave with the deformation recovery of the sample. This part of energy is called the releasable deformation energy $W_e$.

According to the dynamic loading process of jointed rock mass sample, when the stress wave duration ends, that is, $t = t_0$, the loading stress of the sample decreases to 0 and the sample just goes through a dynamic loading and unloading cycle. At this point, the releasable deformation energy $W_e$ of the sample deformation energy has been transferred to the elastic rod in the form of stress wave, and the rest is dissipated energy $W_l$ so $W_l = W_s(t_0)$. Equation (11) can be obtained from Equation (10).

$$W_l = W_s(t_0) = W_i(t_0) - W_r(t_0) - W_t(t_0) \tag{11}$$

When the stress wave interacts with the jointed rock mass sample, that is, when $t < t_0$, the deformation energy $W_s(t)$ of the jointed rock mass sample contains two parts of energy, namely, the released deformation energy $W_e$ and the dissipated energy $W_l$. Its maximum

value $W_s max$ is the maximum deformation energy of the deformation energy during the DYNAMIC test of SHPB, which can be expressed as Equation (12).

$$W_s max = \max_{0 \le t \le t0} W_s(t) = \max_{0 \le t \le t0} \left( AcE \int_0^{t_0} \varepsilon_i^2(t) - \varepsilon_r^2(t) dt - \varepsilon_t^2(t) \right) \tag{12}$$

According to Equations (10) and (11), the dissipated energy of jointed rock mass sample can be calculated, as shown in Equations (13) and (14).

$$W_l = W_s(t_0) = Ac_0 E \int_0^{t_0} \varepsilon_i^2(t) - \varepsilon_r^2(t) dt - \varepsilon_t^2(t) \tag{13}$$

$$Q_s = 2\pi \frac{Wsmax}{W_l} \tag{14}$$

The time–history curves of three-wave strain in Figure 8 are substituted into Equations (12)–(14), and the seismic wave quality factors of jointed rock mass can be obtained with the aid of data processing software.

### 4.2. Acquisition of Seismic Wave Quality Factor of Jointed Rock Mass

The software MATLAB was used to program Equations (12)–(14). Meanwhile, the time–history data of three-wave strain in Figure 8 were imported into the MATLAB workspace to establish the data matrix, and the data matrix was analyzed and processed based on the calculation function obtained by the program. Finally, the maximum storage energy $W_s max$, dissipation energy $W_l$ and seismic wave quality factor $Qs$ of the jointed rock mass under the loading condition of six groups of three specimens were obtained, as shown in Table 1.

**Table 1.** Maximum stored energy. $W_s max$, dissipation energy $W_l$ and seismic wave quality factor $Q_s$ under different loading conditions.

| Specimen Number | Loading Condition | $W_s max$/J | $W_l$/J | $Qs$ |
|---|---|---|---|---|
| 1 | ① | 13.21 | 1.93 | 43.0 |
|   | ② | 11.89 | 1.71 | 43.7 |
| 2 | ① | 13.35 | 1.97 | 42.6 |
|   | ② | 11.57 | 1.72 | 42.3 |
| 3 | ① | 12.94 | 1.88 | 43.2 |
|   | ② | 11.68 | 1.68 | 43.6. |
| **Average** | | 12.44 | 1.815 | 43.07 |

Based on the above analysis, it can be seen that under different loading conditions of the same group of specimens, the measured maximum stored energy, dissipated energy and seismic wave quality factor are relatively close, and the maximum deviation of the quality factor $Qs$ is 3.2%, indicating that the loading conditions of this experiment are set reasonably and the conventional joint cracks are not opened by the first impact. In addition, although the experimental results show some discreteness, the overall deviation is not large and the average $Qs$ is 43.07.

## 5. Numerical Simulation of Damage and Fracture of Rock Mass under Explosive Dynamic Loading

### 5.1. AUTODYN-Code Finite Difference Calculation Principle

AUTODYN-Code is a general program for displaying nonlinear dynamic finite differences using finite difference, finite volume, and finite element techniques to solve a variety of nonlinear problems in solid, fluid, and gas dynamics. The AUTODYN-Code solver firstly divides the solution region of the model into a finite number of difference grids, and then

replaces the continuous solution region with finite grid nodes. Finally, the solution of the difference equation matrix is the numerical solution of the flow variable on the grid nodes.

*5.2. Selection of Numerical Models for Materials and Explosives*

Equation of State (EOS), strength model and failure model of test materials are the three aspects that need to be established in numerical model, which are also the key to improve the matching degree of numerical and physical test. Considering the equation of state, strength model and failure model, a more reasonable numerical model is selected for the test materials and explosives in this paper.

### 5.2.1. Equation of State of Specimen Material

Compared with gas and liquid, the change in internal energy and volume of condensed matter is very small, especially that of rock brittle material. Therefore, the equation of state of rocks and other materials that can be destroyed under small deformation can be expressed in the following Equation (15), namely, linear equation of state Liner:

$$P = k\mu = k\left(\frac{\rho}{\rho_0}\right) - 1 \tag{15}$$

where $P$ is the volume stress, $k$ is the volume modulus, $\rho/\rho_0$ is the ratio of the transient density to the initial state density of the material.

### 5.2.2. Equation of State of Explosive

The numerical model adopts dynamite loading, and its selection will affect the accuracy of simulation to a certain extent. Currently, the widely used explosive equation of state is *JWL*, and its equation of state and isentropic equation are shown in Equations (16) and (17):

$$P = A \cdot \left(1 - \frac{\omega}{R_1 V}\right) \cdot e^{-R_1 V} + B \cdot \left(1 - \frac{\omega}{R_2 V}\right) \cdot e^{-R_2 V} + \frac{\omega E_0}{V} \tag{16}$$

$$P_s = A \cdot e^{-R_1 V} + B \cdot e^{-R_2 V} + CV^{-(\omega+1)} \tag{17}$$

In the Equations (16) and (17), P is the compressive stress, $V = \rho/\rho_0$ is the ratio of the density after explosion to the initial explosive density, and $E_0 = \rho_0 e$ is the initial volume energy. When the explosive type is determined, the parameter explosion velocity and explosion pressure can be obtained. $A$, $B$, $C$, $R_1$, $R_2$ and $\omega$ are undetermined parameters. By assuming a set of $R_1$, $R_2$ and $\omega$ values, $A$, $B$ and $C$ can be obtained from the $CJ$ condition of dynamite, the relation between isentropical line $P_s$ and Rayleigh line and Hugoniot curve passing through $CJ$ point. At this point, $R_1$, $R_2$ and $\omega$ can be obtained through barrel test calibration. The ANFO explosive model is used in numerical simulation in this paper, and the undetermined parameters in Equation (16) are shown as follows:

$$\left.\begin{array}{l} A = 49.46001 \, GPa; \, B = 1.89100 \, GPa \\ R_1 = 3.907, \, R_2 = 1.118, \, D_{cj} = 4160 \, m \cdot s^{-1} \\ \omega = 0.33333, \, E_{cj} = 2.484e^6 \, kj \cdot m^{-1} \\ P_{cj} = 5.15 \, GPa, \, \rho = 0.931 \, g \cdot cm^{-1} \end{array}\right\} \tag{18}$$

The three terms contained in Equation (16), $A \cdot (1 - \omega/R_1 V)$ plays a major role in the high pressure stage of explosion, $B \cdot (1 - \omega/R_2 V)$ in the medium-pressure stage of explosion, and $\omega E_0/V$ in the low-pressure stage of explosion, as shown in Figure 9.

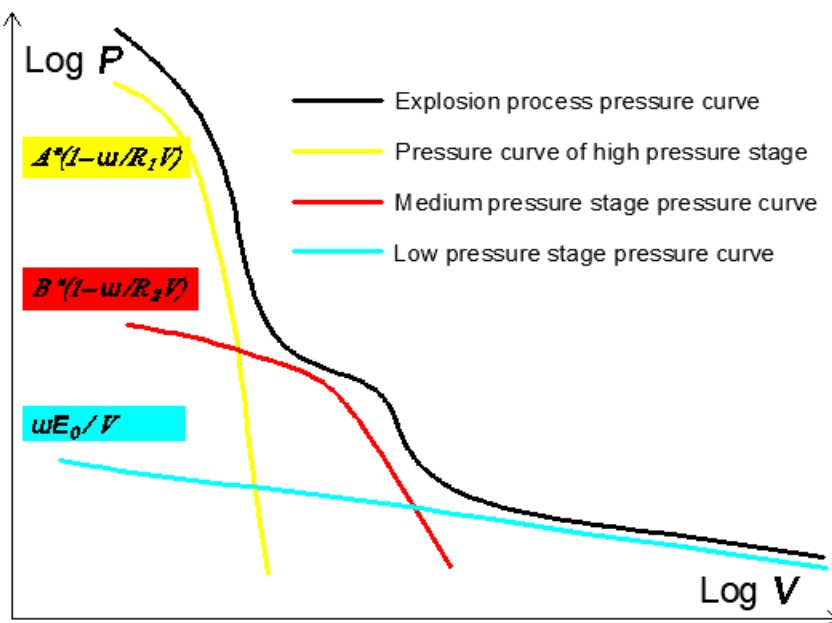

**Figure 9.** JWL equation pressure and relative volume number function diagram.

### 5.2.3. The Intensity of the Model

(1) Strength model of specimen material

Under the condition of static load or quasi static load, brittle materials, such as rock before reach the ultimate strength of the stress–strain curve can be divided into the following stages: pressure dense phase *OA*; elastic deformation stage *AB*; crack development stage *BC*; yield stage *CD*, the sigma $\sigma_E$ for the elastic limit; yield limit for sigma $\sigma_S$; sigma $\sigma_C$ as the ultimate strength, as shown in Figure 10a. When the rock is under high loading rate such as explosion or impact, the original microcracks in the rock undergo a very short compaction stage and then quickly enter the elastic stage. When the rock material enters the elastic deformation stage, due to the high loading rate and transient loading characteristics of the explosion load the rock reaches the strength limit $\sigma_C$ after a very short plastic stage, and then enters the failure stage. The whole process of stress and strain under the dynamic load of the explosion is shown in Figure 10b.

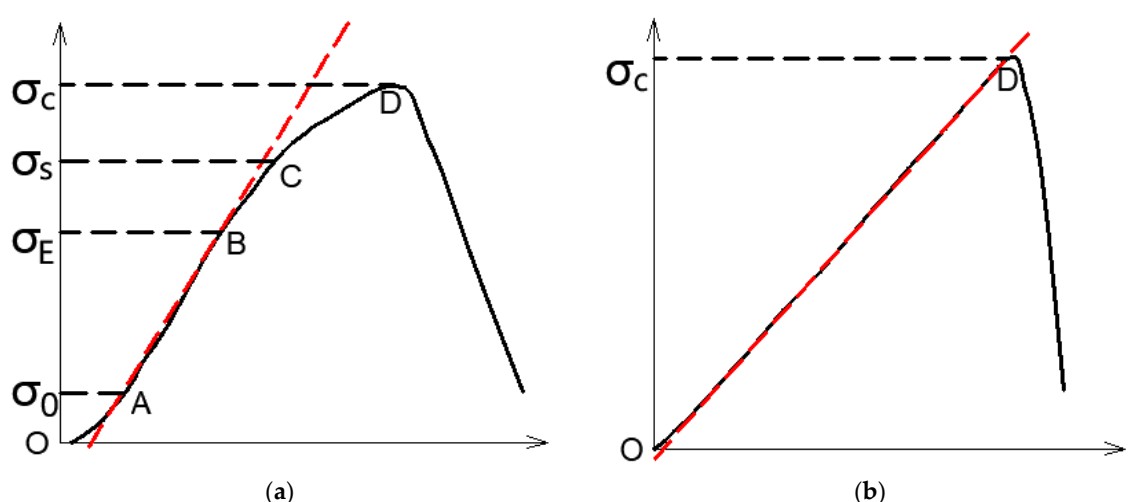

(**a**)    (**b**)

**Figure 10.** Stress–strain curves of rock under static and dynamic loads. (**a**) Static load; (**b**) Dynamic load.

Under high loading rates such as explosion, the linear elastic strength criterion can be adopted for the model of brittle rock materials. Based on the generalized Hu's law, the stress–strain relationship of isotropic elastic model can be expressed as follows:

$$
\begin{bmatrix} \varepsilon_{11} \\ \varepsilon_{22} \\ \varepsilon_{33} \\ \gamma_{12} \\ \gamma_{13} \\ \gamma_{23} \end{bmatrix} = \begin{bmatrix} 1/E & -\mu/E & -\mu/E & 0 & 0 & 0 \\ -\mu/E & 1/E & -\mu/E & 0 & 0 & 0 \\ -\mu/E & -\mu/E & 1/E & 0 & 0 & 0 \\ 0 & 0 & 0 & 1/G & 0 & 0 \\ 0 & 0 & 0 & 0 & 1/G & 0 \\ 0 & 0 & 0 & 0 & 0 & 1/G \end{bmatrix} \begin{bmatrix} \sigma_{11} \\ \sigma_{22} \\ \sigma_{33} \\ \tau_{12} \\ \tau_{13} \\ \tau_{23} \end{bmatrix}
\tag{19}
$$

where $E$ is the elastic model, $\mu$ is Poisson's ratio, and $G$ is shear modulus $G = E/2(1+v)$.

Rocks are condensed porous material. When there is gap stress concentration will occur at the top of the gap resulting in holes in the material, which is prone to brittle fracture. Under high loading rates such as explosion or impact, its fracture morphology will change from static and quasi-static tensile shear composite failure to tensile fracture failure, that is, failure can occur under small deformation conditions. It is similar and consistent with the stress–strain law in fracture behavior of rock brittle materials.

(2) Explosive strength model

The condensed matter explosive will be transformed into explosive gas after the explosive reaction. In the vicinity of the borehole, apart from the stress wave transmission of shock wave, there is still a large amount of explosive gas and the effect of that can be ignored. Therefore, the strength model of explosive used in numerical simulation is None.

5.2.4. The Failure Model

(1) Failure model of specimen material

When the strength of the local stress field of material under the action of external force reaches or exceeds its strength limit, the failure phenomenon will occur. In order to better simulate the dynamic fracture morphology of rock in a physical test, a reasonable failure model should be selected in numerical simulation. In this paper, the modified maximum principal stress criterion is used to describe the fracture failure behavior of rock under explosive dynamic load. In general, the description of the maximum principal stress failure criterion for the material state can also be simplified as: when the principal stress $\sigma_1$ of A certain element reaches the dynamic tensile strength $\sigma_T$ or the maximum shear stress $\tau_{max}$ reaches the dynamic shear strength $\tau_c$ of the material, cracks begin to form and are in a state of damage. When the crack runs through the whole unit, it has completely failed. In this case, $\sigma_1 = 0$ of the unit is described by Equation (20).

$$
\sigma_T \leq \sigma_T \text{ or } \tau_{max} \leq \tau_c
\tag{20}
$$

In combination with the crack tip stress intensity factor and its critical stress criterion Equation in linear elastic fracture mechanics, the relationship between the critical stress of model element and the crack propagation length within the element can be obtained, as shown in Equations (21) and (22).

$$
K = \alpha \cdot \sqrt{\pi a} = K_{IC}^d
\tag{21}
$$

$$
\sigma_1 = \frac{K_{IC}^d}{\alpha \cdot \sqrt{\pi a}} = \frac{\sqrt{E_d G_c}}{\alpha \cdot \sqrt{\pi a}}
\tag{22}
$$

where $K_{IC}^d$ is the material dynamic fracture toughness, $E_d$ is the material dynamic elastic modulus, $a$ is the dynamic crack propagation length, $G_c$ is the energy required per unit thickness model crack dynamic fracture length, and $\alpha$ is the shape coefficient related to crack size and location.

To describe the gradual failure of the element, this paper adopts the maximum principal stress linear tensile fracture softening damage model, and the relationship between the maximum principal stress and strain of the element is shown in Figure 11.

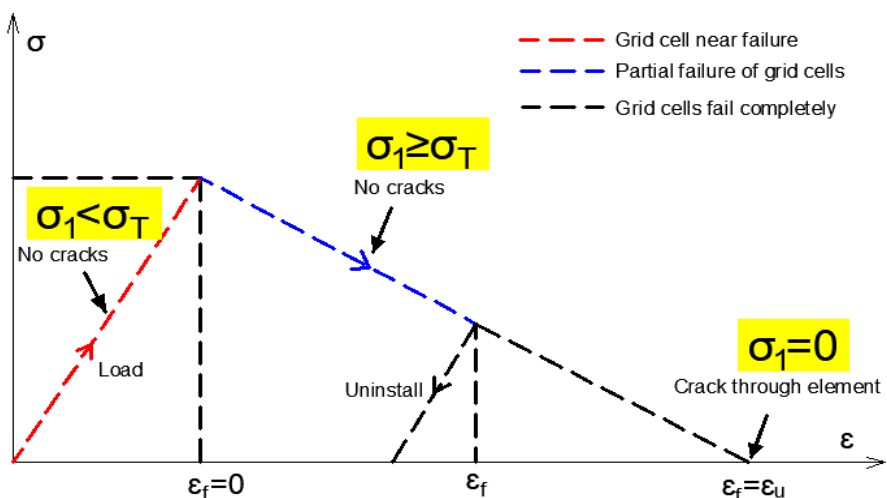

**Figure 11.** Diagram of softening model $\sigma_1 - \varepsilon_F$.

In the fracture softening model, the dynamic fracture strain after complete failure of model element is $\varepsilon_\mu$, the energy required for crack propagation per unit length on specimen with unit thickness is $G_C$, and the dynamic fracture strain $\varepsilon_\mu$ is expressed as Equation (23):

$$\varepsilon_\mu = \frac{2G_C}{\sigma_T L} \tag{23}$$

where $\sigma_T$ is the dynamic tensile strength of the material, and $L$ is the characteristic size in the direction of maximum principal stress.

In addition, the damage variable *Dam* of the model element is the ratio of dynamic fracture strain $\varepsilon_f$ to complete fracture strain $\varepsilon_\mu$.

$$Dam = \frac{\varepsilon_f}{\varepsilon_\mu} = \frac{\varepsilon_f \sigma_T L}{2G_C} \tag{24}$$

In the Equation (24), dynamic fracture strain $\varepsilon_f$ can be obtained by backward Euler method.

Therefore, when the model element is damaged from the beginning to complete failure, the maximum principal stress $\sigma_1$ that it can withstand can be expressed as follows:

$$\sigma_1 = \sigma_T(1 - Dam) \tag{25}$$

According to Equation (25), it can be seen that the whole fracture damage process of model element changes with $Dam = (0 \sim 1)$.

(2) Failure model of explosive

As with the strength model for explosives, the failure model for explosives is None.

### 5.3. Damage and Fracture Analysis of Rock Mass with Different Quality Factors under Explosive Dynamic Load

Based on the above theoretical analysis, nonlinear dynamic finite difference software AUTODYN-Code was used to analyze the damage and breakage of different quality factors under dynamic explosion loading. The state equation of the model is linear, the strength model is linear elastic, and the failure model is linear tensile fracture softening damage model of maximum principal stress. The state equation of explosive is *JWL* state equation, the strength model is None, and the failure model of explosive is None.

The simulated size diameter of the model is 50 mm, the aperture radius is 2 mm, and the cartridge radius is 1.2 mm. Because engineering rock mass is infinite rock mass, when the explosion shock wave generated after explosive explosion propagates in rock mass it will gradually attenuate to stress wave, and finally attenuate to sound wave and dissipate, without wave reflection. The model adopts Transmit boundary condition, and when the explosion stress wave propagates to the boundary it will spread out and will not form reflected tension wave at the boundary. Therefore, Transmit boundary conditions can better simulate the propagation characteristics of stress wave in rock mass explosion. Figure 12a–f shows the process from the moment of detonation of the explosive near the hole to the propagation of explosive shock wave.

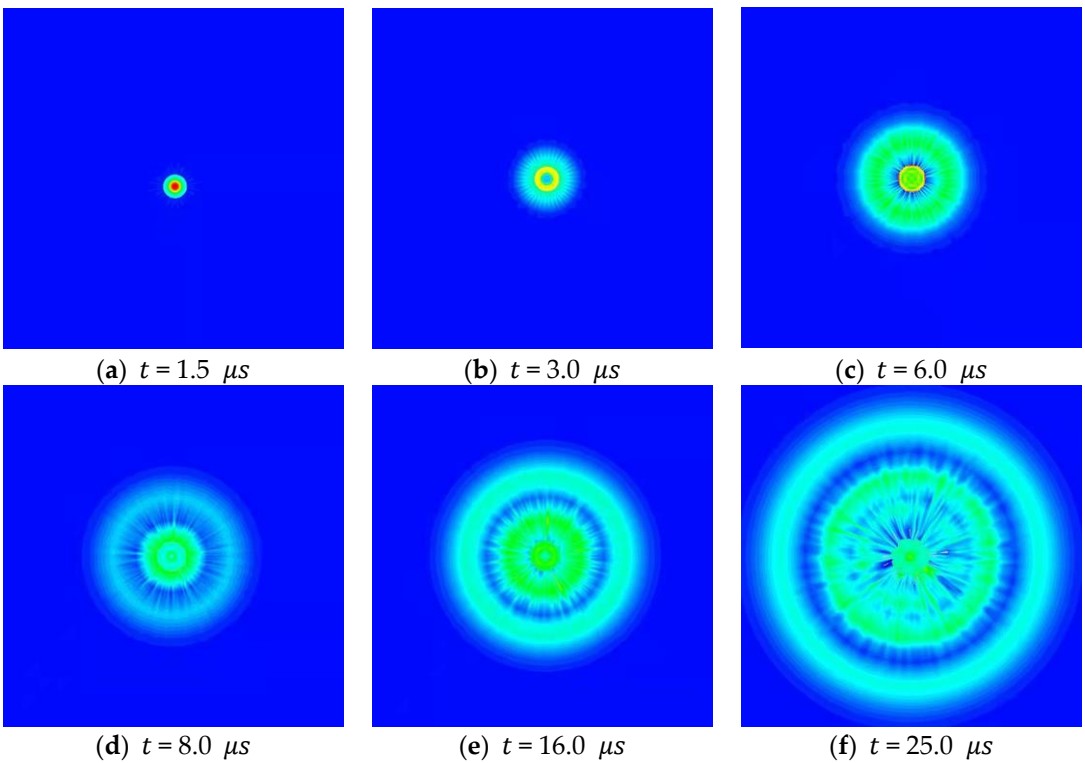

(**a**) $t = 1.5 \ \mu s$     (**b**) $t = 3.0 \ \mu s$     (**c**) $t = 6.0 \ \mu s$

(**d**) $t = 8.0 \ \mu s$     (**e**) $t = 16.0 \ \mu s$     (**f**) $t = 25.0 \ \mu s$

**Figure 12.** Effect diagram of stress wave propagation after detonation.

The seismic wave quality factor of rock mass is the ratio of the maximum variable performance to the dissipated energy when the seismic wave propagates in the rock mass, and it is a dimensionless parameter. The AUTODYN-Code is used to carry out numerical simulation, and the rock mass is used as a porous medium. By setting the numerical model rock mass porosity, the damage and fracture state of the rock mass with different quality factors under the dynamic load of explosion can be simulated by weakening the parameters, as shown in Table 2. Adjust the porosity of the rock mass to simulate the damage and fracture characteristics of the rock mass with different quality factor models under dynamic explosion loading, as shown in Figure 13.

**Table 2.** Different porosity numerical simulation parameters.

| Porosity/% | Model Size/mm | Hole Radius/mm | Radius of Cartridge/mm | Equation of State | Strength Model | Failure Model |
|---|---|---|---|---|---|---|
| 0.1 | 25 | 2 | 1.2 | JWL | None | None |
| 0.15 | 25 | 2 | 1.2 | JWL | None | None |
| 0.18 | 25 | 2 | 1.2 | JWL | None | None |
| 0.25 | 25 | 2 | 1.2 | JWL | None | None |

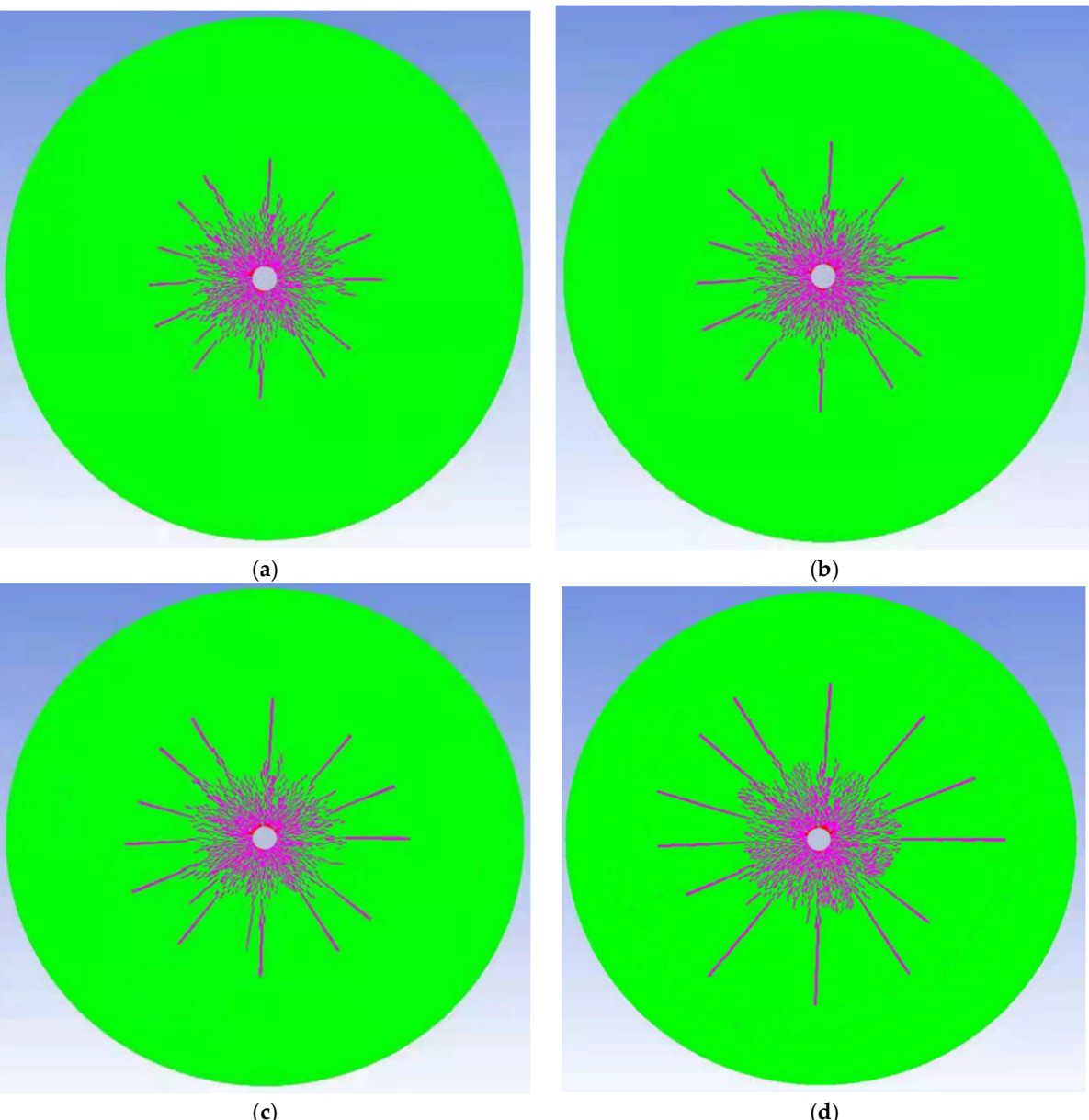

**Figure 13.** Damage and fracture characteristics of rock mass with different porosity under explosive dynamic load. (**a**) The porosity is 0.1%; (**b**) The porosity is 0.15%; (**c**) The porosity is 0.18%; (**d**) The porosity is 0.25%.

Figure 13 shows that the fracture range of the rock mass increases gradually with the increase in the rock mass porosity under the same explosive dynamic load conditions. The rock mass near the blasthole presents a compression–shear crushing zone under the ultra-high loading rate of the shock wave. The four fracture renderings show that the radius of the compression–shear damage zone increases gradually with the increase in porosity, but it is not obvious. The porous media properties of rock mass cause the energy of the blast shock wave to decay exponentially in the range of three to seven times the coil, and then decay into a stress wave. Under the action of attenuated stress wave, the fractures of the surrounding rock mass show tensile fracture characteristics, and no shear fracture cracks appear. When the energy release rate of the stress wave at the tip of the dynamic expanding crack is less than the energy dissipation rate required for the crack expansion, the dynamic expanding crack will stop expanding.

### 6. Conclusions and Prospect

(1) The SHPB dynamic test system was used to obtain the non-damage fracture time–history strain curve of the marble test under impact load; combined with the calculation principle of the deformation energy and dissipation energy of the quality factor, six groups of *Qs* experimental values are obtained. There is a certain discreteness, but the overall deviation is not large with a mean value of 43.07;

(2) Using AUTODYN-Code to simulate the damage and fracture characteristics of different quality factor models of rock mass under dynamic explosion load, the results show that the radius of the rock mass compression shear damage area gradually increases with the increase in porosity, but it is not obvious;

(3) The porous media properties of rock mass cause the energy of the blast shock wave to decay exponentially in the range of three to seven times the coil, and then decay into a stress wave;

(4) Under the action of attenuated stress waves, the fractures of the surrounding rock mass show tensile fracture characteristics, and no shear fracture cracks appear. When the energy release rate of the stress wave at the tip of the dynamic expanding crack is less than the energy dissipation rate required for the crack expansion, the dynamic expanding crack will stop expanding;

(5) This paper only studies the strain values of non-damage fracture history of marble mass test and simulates the damage and fracture characteristics of rock mass model with different quality factors under explosive dynamic load. It is suggested to add the study of rock stability after blasting damage and fracture in the next research.

**Author Contributions:** Conceptualization, C.S.; methodology, C.L. and X.W.; writing—original draft preparation, C.S. All authors have read and agreed to the published version of the manuscript.

**Funding:** This research received no external funding.

**Institutional Review Board Statement:** Not applicable for studies not involving humans or animals.

**Informed Consent Statement:** Not applicable.

**Data Availability Statement:** The study did not report any data.

**Conflicts of Interest:** The authors declare no conflict of interest.

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
