# Peer review of "Research on Seismic Wave Quality Factor of Marble Jointed Rock Mass under SHPB Impact"

_applsci, doi:10.3390/app122110875_

Round 1

Reviewer 1 Report

There are some weaknesses through the manuscript which need improvement. Therefore, the submitted manuscript cannot be accepted for publication in this form, but it has a chance of acceptance after a major revision. My comments and suggestions are as follows:

1- Abstract gives information on the main feature of the performed study, but a couple of sentences about experimental practices must be added.

2- Authors must clarify necessity of the performed research. Research questions, aims and objectives of the study must be clearly mentioned in introduction.

3- The literature study must be enriched. In this respect, authors must read and refer to the following papers: (a) https://doi.org/10.1016/j.apples.2021.100043 (b) https://doi.org/10.1016/j.ijimpeng.2019.103336

4- All figures must be presented in a high quality (e.g., text in Fig. 1 is illegible).

5- There are many discussions and arguments without proper references. Also, some ways of citation are improper.

6- It seems the manuscript is prepared without care. For example, font size must be the same throughout the manuscript. Center align formula must be obeyed. Time/us in Fig. 8 is not correct (microsecond is correct), and so on.

7- Dynamic equilibrium is an important issue in test vis Hopkinson bar. Authors must clarify how dynamic equilibrium is achieved? In this context, they can read and refer to https://doi.org/10.1016/j.tafmec.2017.10.001 and https://doi.org/10.1016/j.ijimpeng.2021.104109

8- Input parameters of numerical simulation must be summarized in a table.

9- Limitations of the study must be discussed.

10- Standard deviation in presented results (curves) must be discussed.

11- In its language layer, the manuscript should be considered for English language editing. There are sentences which have to be rewritten.

12- The conclusion must be more than just a summary of the manuscript. List of references must be updated based on the proposed papers. Please provide all changes by red color in the revised version.

Author Response

Response to Reviewer 1 Comments

Point 1: Abstract gives information on the main feature of the performed study, but a couple of sentences about experimental practices must be added.

Response 1: Added in the abstract: Based on the basic concept of seismic wave quality factor, a method is proposed to calculate seismic wave quality factor of sample by the stress wave energy propagating in elastic rod, and the method is verified theoretically and experimentally. (in red)

Point 2: Authors must clarify necessity of the performed research. Research questions, aims and objectives of the study must be clearly mentioned in introduction.

Response 2: Added in the introduction: In this paper, in order to control blasting vibration and prevent its threat to adjacent structures, it is necessary to deeply understand the propagation law of stress wave in rock mass. The marble from the roof and floor of Jinchanghe Lead-zinc mine in Baoshan was used as test material.  (in red)

Point 3: The literature study must be enriched. In this respect, authors must read and refer to the following

papers:  (a) https://doi.org/10.1016/j.apples.2021.100043

 (b) https://doi.org/10.1016/j.ijimpeng.2019.103336

Response 3: Added in the introduction:Kong xiangzhen[25] researched that the nonlocal model based on damage can solve all the limitations of the original nonlocal model. Mohammad Reza Khosravani[26] used the Holmquist-Johnson-Cook constitutive model to characterize the dynamic behavior of ultra-high performance concrete, which is a new generation of concrete with higher strength compared with traditional concrete. The reflected and transmitted waves obtained from experiments are used as the input in the inversion process.

Point 4: All figures must be presented in a high quality (e.g., text in Fig. 1 is illegible).

Response 4: It has been modified as required. And, after the modification has been tested, print out completely see clearly.

Point 5: There are many discussions and arguments without proper references. Also, some ways of citation are improper.

Response 5: It has been modified as much as possible.

Point 6: It seems the manuscript is prepared without care. For example, font size must be the same throughout the manuscript. Center align formula must be obeyed. Time/us in Fig. 8 is not correct (microsecond is correct), and so on.

Response 6: It has been modified as required.

Point 7: Dynamic equilibrium is an important issue in test vis Hopkinson bar. Authors must clarify how dynamic equilibrium is achieved? In this context, they can read and refer to https://doi.org/10.1016/j.tafmec.2017.10.001 and https://doi.org/10.1016/j.ijimpeng.2021.104109

Response 7: Added in the introduction: F.LI[27] researched that the increase of compressive strength in SHPB test was largely caused by the change of stress state in the sample, that is, when the strain rate was greater than the transition strain rate, the transverse limit was introduced, that is, from uniaxial stress state to multiaxial stress state.

Point 8: Input parameters of numerical simulation must be summarized in a table.

Response8: It has been modified as required. Added in Section 4.3:Table 2. Different porosity numerical simulation parameters

Point 9: Limitations of the study must be discussed.

Response 9: In Section 5, the conclusion is changed into conclusion and prospect. Added in Section 5: (3) The porous media properties of rock mass cause the energy of the blast shock wave to decay exponentially in the range of 3 to 7 times the coil, and then decay into a stress wave.

(4) Under the action of attenuated stress wave, the fractures of the surrounding rock mass show tensile fracture characteristics, and no shear fracture cracks appear. When the energy release rate of the stress wave at the tip of the dynamic expanding crack is less than the energy dissipation rate required for the crack expansion, the dynamic expanding crack will stop expanding.

(5) This paper only studies the strain values of non-damage fracture history of marble mass test, and simulates the damage and fracture characteristics of rock mass model with different quality factors under explosive dynamic load. It is suggested to add the study of rock stability after blasting damage and fracture in the next research.

Point 10: Standard deviation in presented results (curves) must be discussed.

Response 10:  Added in Section 3.2:Based on the above analysis, it can be seen that under different loading conditions of the same group of specimens, the measured maximum stored energy, dissipated energy and seismic wave quality factor are relatively close, and the maximum deviation of quality factor  is 3.2%, indicating that the loading conditions of this experiment are set reasonably, and the conventional joint cracks are not opened by the first impact.

Point 11: In its language layer, the manuscript should be considered for English language editing. There are sentences which have to be rewritten.

Response 11: It has been modified as much as possible.

Point 12: The conclusion must be more than just a summary of the manuscript. List of references must be updated based on the proposed papers. Please provide all changes by red color in the revised version.

Response 12: It has been modified as required. Added in Section 5: (3) The porous media properties of rock mass cause the energy of the blast shock wave to decay exponentially in the range of 3 to 7 times the coil, and then decay into a stress wave.

(4) Under the action of attenuated stress wave, the fractures of the surrounding rock mass show tensile fracture characteristics, and no shear fracture cracks appear. When the energy release rate of the stress wave at the tip of the dynamic expanding crack is less than the energy dissipation rate required for the crack expansion, the dynamic expanding crack will stop expanding.

(5) This paper only studies the strain values of non-damage fracture history of marble mass test, and simulates the damage and fracture characteristics of rock mass model with different quality factors under explosive dynamic load. It is suggested to add the study of rock stability after blasting damage and fracture in the next research.

Add in section 6: [25] Kong Xiangzhen, Fang Qin, Hong Jian. A new damage-based nonlocal model for dynamic tensile failure of concrete material[J]. International Journal of Impact Engineering, 2019, 132(10): 103336.

[26] Mohammad Reza Khosravani. Inverse characterization of UHPC material based on Hopkinson bar test[J]. Applications in Engineering Science, 2021,(6):1-11.

[27] F.LI, Q.M.LI. Strain-rate effect of polymers and correction methodology in a SHPB test[J]. International Journal of Impact Engineering, 2022, 161(10):104109.

Reviewer 2 Report

In this paper, SHPB test system is used to conduct impact compression tests on multiple samples of the same rock mass. A quality factor is introduced to measure the dissipation of stress waves in jointed marble rock masses. The AUTODYN code is used to simulate the damage and fracture characteristics of rock mass with different quality factors under explosive dynamic load. The research objective and conclusion of this paper are clear, the research content is rich, and the research method is novel, which is of great significance to the evaluation of rock properties.

1.In Section 1.4, please supplement  and under static conditions.

2.In Section 2.2, please supplement and  under dynamic conditions. Introduce the relationship or calculation formula between them.

3. In the abstract and conclusions of this paper, please write down the specific groups of "several sets of  empirical values were observed", for example, three groups, four groups or five groups.

4.It is suggested to add prospects for further research in this field.

5. Some newest relevant publications should be cited.

Author Response

Response to Reviewer 2 Comments

Point 1: In Section 1.4, please supplement  and under static conditions.

Response 1: Added in section 1.4: The parameters of the processed specimens were measured by the sound velocimeter, and the compression wave velocity Cp=3731m/s and shear wave velocity Cs=2308m/s were obtained.

Point 2: In Section 2.2, please supplement and  under dynamic conditions. Introduce the relationship or calculation formula between them.

Response 2: Added in section 2.2:The fundamental frequency of the specimen was obtained by using a dynamometer, and the dynamic elastic modulus of the specimen was 34.68g by combining the fundamental frequency parameters with the wave velocity measured by a sonic meter. The dynamic Poisson's ratio =0.26 can be obtained by combining elastic wave theory with Poisson's ratio Eq..

But no correlation formula was found.

Point 3: In the abstract and conclusions of this paper, please write down the specific groups of "several sets of  empirical values were observed", for example, three groups, four groups or five groups.

Response 3: It has been modified as required. Add in the Abstract and Conclusions: The results show that: ① The non-destructive breaking time history strain of Dali rock mass under impact load is obtained by SHPB dynamic test system; Combined with the deformation energy and dissipation energy calculation principle of quality factor, six groups of  experimental values are obtained. Although the  experimental values are discrete, the overall deviation is small, with an average of 43.07.

Point 4: It is suggested to add prospects for further research in this field.

Response 4: It has been modified as required. In Section 5, it is changed to Conclusion and Prospect,add in scetion5: (3) The porous media properties of rock mass cause the energy of the blast shock wave to decay exponentially in the range of 3 to 7 times the coil, and then decay into a stress wave.

(4) Under the action of attenuated stress wave, the fractures of the surrounding rock mass show tensile fracture characteristics, and no shear fracture cracks appear. When the energy release rate of the stress wave at the tip of the dynamic expanding crack is less than the energy dissipation rate required for the crack expansion, the dynamic expanding crack will stop expanding.

(5) This paper only studies the strain values of non-damage fracture history of marble mass test, and simulates the damage and fracture characteristics of rock mass model with different quality factors under explosive dynamic load. It is suggested to add the study of rock stability after blasting damage and fracture in the next research.

Point 5: Some newest relevant publications should be cited.

Response 5: It has been modified as required.

 Add in the introduction:

 At present, most of the researches focus on the amplitude change of stress wave after it passes through the joint, and the energy attenuation of stress wave is rarely analyzed, stress wave energy attenuation is an important part of stress wave propagation law, which can provide a certain reference for blasting vibration analysis and prediction[12-13]. Therefore, it is valuable to deeply understand the influence of joints on stress wave energy attenuation in rock mass.

Kong xiangzhen[25] researched that the nonlocal model based on damage can solve all the limitations of the original nonlocal model. Mohammad Reza Khosravani[26] used the Holmquist-Johnson-Cook constitutive model to characterize the dynamic behavior of ultra-high performance concrete, which is a new generation of concrete with higher strength compared with traditional concrete. The reflected and transmitted waves obtained from experiments are used as the input in the inversion process. F.LI[27] researched that the increase of compressive strength in SHPB test was largely caused by the change of stress state in the sample, that is, when the strain rate was greater than the transition strain rate, the transverse limit was introduced, that is, from uniaxial stress state to multiaxial stress state.

Add in section 6:

[12] Cai X, Cheng C, Zhao Y, Zhou Z, Wang S. The role of water content in rate dependence of tensile strength of a fine-grained sandstone[J]. Archives of Civil and Mechanical Engineering, 2022, 22(1): 1-16.

[13] Zhou Z, Cai X, Li X, Cao W, Du X. Dynamic Response and Energy Evolution of Sandstone Under Coupled Static–Dynamic Compression: Insights from Experimental Study into Deep Rock Engineering Applications[J]. Rock Mechanics and Rock Engineering, 2020, 53: 1305-1331.

[25] Kong Xiangzhen, Fang Qin, Hong Jian. A new damage-based nonlocal model for dynamic tensile failure of concrete material[J]. International Journal of Impact Engineering, 2019, 132(10): 103336.

[26] Mohammad Reza Khosravani. Inverse characterization of UHPC material based on Hopkinson bar test[J]. Applications in Engineering Science, 2021,(6):1-11.

[27] F.LI, Q.M.LI. Strain-rate effect of polymers and correction methodology in a SHPB test[J]. International Journal of Impact Engineering, 2022, 161(10):104109.

Round 2

Reviewer 1 Report

The paper has been improved and corresponding modifications have been conducted. In my opinion, the current version can be considered for publication. 

Author Response

1、The references were sorted out and added

[28]WANG Chunlai, LIU Yubo, HOU Xiaolin, DavideElmo. Investigation of the spatial distribution pattern of 3D microcracks in single-cracked breakage[J]. International Journal of Rock Mechanics and Mining Sciences, 2022,154(6):105126.

[29] WANG Chunlai, ZHOU Baokun, LI Changfeng, et al. Experimental investigation on the spatio-temporal-energy evolution pattern of limestone fracture using acoustic emission monitoring[J]. Journal of Applied Geophysics, 2022,206(11):104787.

[30] LI Xuelong, CHEN Shaojie, WANG Enyan, Li Zhouhui. Rockburst mechanism in coal rock with structural surface and the microseismic (MS) and electromagnetic radiation (EMR) response[J]. Engineering Failure Analysis, 2021,124:105396.

[31] FANG Fan, CHEN Shaojie, WANG Yajun, et al. Cracking mechanism and strength criteria evaluation of granite affected by intermediate principal stresses subjected to unloading stress state[J]. International Journal of Rock Mechanics and Mining Sciences,2021,143:104783.

2、The pictures were annotated
